# Reinforcement Learning with Neural Radiance Fields

**Danny Driess**[*]  **Ingmar Schubert**[*]  **Pete Florence**  **Yunzhu Li**  **Marc Toussaint**
TU Berlin        TU Berlin        Google        MIT        TU Berlin

## Abstract

It is a long-standing problem to find effective representations for training reinforcement learning (RL) agents. This paper demonstrates that learning state representations with supervision from Neural Radiance Fields (NeRFs) can improve the performance of RL compared to other learned representations or even low-dimensional, hand-engineered state information. Specifically, we propose to train an encoder that maps multiple image observations to a latent space describing the objects in the scene. The decoder built from a latent-conditioned NeRF serves as the supervision signal to learn the latent space. An RL algorithm then operates on the learned latent space as its state representation. We call this NeRF-RL. Our experiments indicate that NeRF as supervision leads to a latent space better suited for the downstream RL tasks involving robotic object manipulations like hanging mugs on hooks, pushing objects, or opening doors.
Video: https://dannydriess.github.io/nerf-rl

## 1 Introduction

The sample efficiency of reinforcement learning (RL) algorithms crucially depends on the representation of the underlying system state they operate on [1, 2, 3, 4, 5, 6, 7]. Sometimes, a low-dimensional (direct) representation of the state, such as the positions of the objects in the environment, is considered to make the resulting RL problem most efficient [2].

However, such low-dimensional, direct state representations can have several disadvantages. On the one hand, a perception module, e.g., pose estimation, is necessary in the real world to obtain the representation from raw observations, which often is difficult to achieve in practice with sufficient robustness. On the other hand, if the goal is to learn policies that generalize over different object shapes [8], using a low-dimensional state representation is often impractical. Such scenarios, while challenging for RL, are common, e.g., in robotic manipulation tasks.

Therefore, there is a large history of approaches that consider RL directly from raw, high-dimensional observations like images (e.g., [9, 10]). Typically, an encoder takes the high-dimensional input and maps it to a low-dimensional latent representation of the state. The RL algorithm (e.g., the Q-function or the policy network) then operates on the latent vector as state input. This way, no separate perception module is necessary, the framework can extract information from the raw observations that are relevant for the task, and the RL agent, in principle, may generalize over challenging environments, in which, e.g., object shapes are varied. While these are advantages in principle, jointly training encoders capable of processing high-dimensional inputs from the RL signal alone is challenging. To address this, one approach is to *pretrain* the encoder on a different task, e.g., image reconstruction [1, 4, 11], multi-view consistency [6], or a time-constrastive task [3]. Alternatively, an auxiliary loss on the latent encoding can be added *during* the RL procedure [5].

In both cases, the choice of the actual (auto-)encoder architecture and associated (auxiliary) loss function has a significant influence on the usefulness of the resulting latent space for the downstream

---

[*]equal contribution. Correspondence: danny.driess@gmail.com

36th Conference on Neural Information Processing Systems (NeurIPS 2022).

RL task. Especially for image data, convolutional neural networks (CNNs) are commonly used for the encoder [12]. However, 2D CNNs have a 2D (equivariance) bias, while for many RL tasks, the 3D structure of our world is essential. Architectures like Vision Transformers [13, 14] process images with no such direct 2D bias, but they often require large scale data, which might be challenging in RL applications. Additionally, although multiple uncalibrated 2D image inputs can be used with generic image encoders [15], they do not benefit from 3D inductive biases, which may help for example in resolving ambiguities in 2D images such as occlusions and object permanence.

Recently, Neural Radiance Fields (NeRFs) [16] have shown great success in learning to represent scenes with a neural network that enables to render the scene from novel viewpoints, and have sparked broad interest in computer vision [17]. NeRFs exhibit a strong 3D inductive bias, leading to better scene reconstruction capabilities than methods composed of generic image encoders (e.g., [18]).

In the present work, we investigate whether incorporating these 3D inductive biases of NeRFs into learning a state representation can benefit RL. Specifically, we propose to train an encoder that maps multiple RGB image views of the scene to a latent representation through an auto-encoder structure, where a (compositional) NeRF decoder provides the self-supervision signal using an image reconstruction loss for each view.

In the experiments, we show for multiple environments that supervision from NeRF leads to a latent representation that makes the downstream RL procedure more sample efficient compared to supervision via a 2D CNN decoder, a contrastive loss on the latent space, or even hand-engineered, perfect low-level state information given as keypoints. Commonly, RL is trained on environments where the objects have the same shape. Our environments include hanging mugs on hooks, pushing objects on a table, and a door opening scenario. In all of these, the objects' shapes are not fixed, and we require the agent to generalize over all shapes from a distribution.

To summarize our main contributions: (i) we propose to train state representations for RL with NeRF supervision, and (ii) we empirically demonstrate that an encoder trained with a latent-conditioned NeRF decoder, especially with an object-compositional NeRF decoder, leads to increased RL performance relative to standard 2D CNN auto-encoders, contrastive learning, or expert keypoints.

## 2 Related Work

**Neural Scene/Object Representations in Computer Vision, and Applications.** To our knowledge, the present work is the first to explore if neural scene representations like NeRFs can benefit RL. Outside of RL, however, there has been a very active research field in the area of neural scene representations, both in the representations themselves [19, 20, 21, 22] and their applications; see [23, 24, 17] for recent reviews. Within the family of NeRFs and related methods, major thrusts of research have included: improving modeling formulations [25, 26], modeling larger scenes [26, 27], addressing (re-)lighting [28, 29, 30], and an especially active area of research has been in improving speed, both of training and of inference-time rendering [31, 32, 33, 34, 35, 36, 37, 38, 39, 40, 41]. In our case, we are not constrained by inference-time computation issues, since we do not need to render images, and only have to run our latent-space encoder (with a runtime of approx. 7 ms on an RTX3090). Additionally of particular relevance, various methods have developed latent-conditioned [42, 43, 44] or compositional/object-oriented approaches for NeRFs [45, 46, 47, 48, 49, 50, 51, 52, 53], although they, nor other NeRF-style methods to our knowledge, have been applied to RL. Neural scene representations have found application across many fields (i.e., augmented reality and medical imaging [54]) and both NeRFs [55, 56, 57, 58] and other neural scene approaches [59, 60, 61, 62] have started to be used for various problems in robotics, including pose estimation [55], trajectory planning [56], visual foresight [11, 53], grasping [59, 57], and rearrangement tasks [60, 61, 58].

**Learning State Representations for Reinforcement Learning.** One of the key enabling factors for the success of deep RL is its ability to find effective representations of the environment from high-dimensional observation data [10, 63]. Extensive research has gone into investigating different ways to learn better state representations using various auxiliary objective functions. Contrastive learning is a common objective and has shown success in unsupervised representation learning in computer vision applications [64, 65]. Researchers built upon this success and have shown such learning objectives can lead to better performance and sample efficiency in deep RL [66, 67], where the contrasting signals could come from time alignment [68, 3], camera viewpoints [69], and different sensory modalities [70], with applications in real-world robotic tasks [6, 71]. Extensive efforts

have investigated the role of representation learning in RL [72], provided a detailed analysis of the importance of different visual representation pretraining methods [73], and shown how we can improve training stability in the face of multiple auxiliary losses [74]. There is also a range of additional explorations on pretraining methods with novel objective functions (e.g., bisimulation metrics [75] and temporal cycle-consistency loss [76]) and less-explored data sources (e.g., in-the-wild images [77] and action-free videos [78]). Please check the survey for more related work in this direction [79]. Our method is different in that we explicitly utilize a decoder that includes strong 3D inductive biases provided by NeRFs, which we empirically show improves RL for tasks that depend on the geometry of the objects.

## 3 Background

### 3.1 Reinforcement Learning

This work considers decision problems that can be described as discrete-time Markov Decision Processes (MDPs) $M = \langle \mathcal{S}, \mathcal{A}, T, \gamma, R, P_0 \rangle$. $\mathcal{S}$ and $\mathcal{A}$ are the sets of all states and actions, respectively. The transition probability (density) from $s$ to $s'$ using an action $a$ is $T(s' \mid s, a)$. The agent receives a real-valued reward $R(s, a, s')$ after each step. The discount factor $\gamma \in [0, 1)$ trades off immediate and future rewards. $P_0 : \mathcal{S} \to \mathbb{R}_0^+$ is the distribution of the start state. RL algorithms try to find the optimal policy $\pi^* : \mathcal{S} \times \mathcal{A} \to \mathbb{R}_0^+$, where $\pi^* = \mathrm{argmax}_\pi \sum_{t=0}^\infty \gamma^t \mathbb{E}_{s_{t+1} \sim T(\cdot|s_t, a_t),\, a_t \sim \pi(\cdot|s_t), s_0 \sim P_0} \left[ R(s_t, a_t, s_{t+1}) \right]$. Importantly, in this work, we consider RL problems where the state $s$ encodes both the position and the shape of the objects in the scene. We require the RL agent to generalize over all of these shapes at test time. We can therefore think of the state as a tuple $s = (s_p, s_s)$, where $s_p$ encodes positional information, and $s_s$ encodes the shapes involved. We focus the experiments on sparse reward settings, meaning $R(s, a, s') = R_0 > 0$ for $s' \in \mathcal{S}_g$ and $R(s, a, s') = 0$ for $s \in \mathcal{S} \backslash \mathcal{S}_g$, where the volume of $\mathcal{S}_g \subset \mathcal{S}$ is much smaller than the volume of $\mathcal{S}$. The state space $\mathcal{S}$ usually is low-dimensional or a minimal description of the degrees of freedom of the system. In this work, we consider that the RL algorithm has only access to a (high-dimensional) observation $y \in \mathcal{Y}$ of the scene (e.g., RGB images). In particular, this means that the policy has observations as input $a \sim \pi(\cdot \mid y)$. Since we assume that the underlying state $s = (s_p, s_s)$ is fully observable from $y$, we can treat $y$ like a state for an MDP.

**Reinforcement Learning with Learned Latent Scene Representations.** The general idea of RL with learned latent scene representations is to learn an *encoder* $\Omega$ that maps an observation $y \in \mathcal{Y}$ to a $k$-dimensional *latent vector* $z = \Omega(y) \in \mathcal{Z} \subset \mathbb{R}^k$ of the scene. The actual RL components, e.g., the Q-function or policy, then operate on $z$ as its state description. For a policy $\pi$, this means that the action $a \sim \pi(\cdot \mid z) = \pi(\cdot \mid \Omega(y))$ is conditional on the latent vector $z$ instead of the observation $y$ directly. The dimension $k$ of the latent vector is typically (much) smaller than that of the observation space $\mathcal{Y}$, but larger than that of the state space $\mathcal{S}$.

### 3.2 Neural Radiance Fields (NeRFs)

The general idea of NeRF, originally proposed by [16], is to learn a function $f = (\sigma, c)$ that predicts the emitted RGB color value $c(x) \in \mathbb{R}^3$ and volume density $\sigma(x) \in \mathbb{R}_{\geq 0}$ at any 3D world coordinate $x \in \mathbb{R}^3$. Based on $f$, an image from an arbitrary view and camera parameters can be rendered by computing the color $C(r) \in \mathbb{R}^3$ of each pixel along its corresponding camera ray $r(\alpha) = r(0) + \alpha d$ through the volumetric rendering relation

$$C(r) = \int_{\alpha_n}^{\alpha_f} T_f(r, \alpha) \sigma(r(\alpha)) c(r(\alpha)) \, \mathrm{d}\alpha \qquad \text{with} \qquad T_f(r, \alpha) = \exp\left( -\int_{\alpha_n}^{\alpha} \sigma(r(u)) \, \mathrm{d}u \right). \quad (1)$$

Here, $r(0) \in \mathbb{R}^3$ is the camera origin, $d \in \mathbb{R}^3$ the pixel dependent direction of the ray and $\alpha_n, \alpha_f \in \mathbb{R}$ the near and far bounds within which objects are expected, respectively. The camera rays are determined from the camera matrix $K$ (intrinsics and extrinsics) describing the desired view.

## 4 Learning State Representations for RL with NeRF Supervision

This section describes our proposed framework, in which we use a latent state space for RL that is learned from NeRF supervision. For learning the latent space, we use an encoder-decoder where the

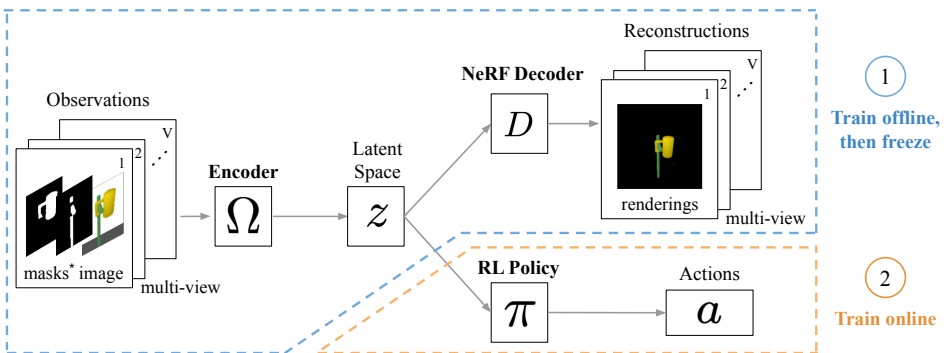

Figure 1: State representation learning for RL with NeRFs. First, the encoder and NeRF decoder are trained with supervision from a multi-view reconstruction loss on an offline dataset. Then, the encoder's weights are frozen, and the latent space is used as state input to train a policy with RL. *Masks of individual objects are only required for the compositional variant of our encoder.

decoder is a latent-conditioned NeRF, which may either be a global [42, 43, 44] or a compositional NeRF decoder [53]. To our knowledge, no prior work has used such NeRF-derived supervision for RL. In Sec. 4.1 we describe this proposition, Sec. 4.2 provides an overview of the encoder-decoder training, Sec. 4.3 and Sec. 4.4 introduce options for the NeRF decoder and encoder, respectively.

## 4.1  Using Latent-Conditioned NeRF for RL

We propose the state representation $z$ on which an RL algorithm operates to be a latent vector produced by an encoder that maps images from multiple views to a latent $z$, which is trained with a (compositional) latent-conditioned NeRF decoder. As will be verified in experiments, we hypothesize that this framework is beneficial for the downstream RL task, as it produces latent vectors that represent the actual 3D geometry of the objects in the scene, can handle multiple objects well, as well as fuse multiple views in a consistent way to deal with occlusions by providing shape completion, all of which is relevant to solve tasks where the geometry is important. There are two steps to our framework, as shown in Fig. 1. First, we train the encoder + decoder from a dataset collected by random interactions with the environment, i.e., we do not yet need a trained policy. Second, we take the encoder trained in the first step, which we leave frozen, and use the latent space to train an RL policy. Note that we investigate two variants of the auto-encoder framework, a global one, where the whole scene is represented by one single latent vector, and a compositional one, where objects are represented by their own latent vector. For the latter, objects are identified by masks in the views.

## 4.2  Overview: Auto-Encoder with Latent-Conditioned NeRF Decoder

Assume that an observation $y = \left( I^{1:V}, K^{1:V}, M^{1:V} \right)$ of the scene consists of RGB images $I^i \in \mathbb{R}^{3 \times h \times w}$, $i = 1, \ldots, V$ taken from $V$ many camera views, their respective camera projection matrices $K^i \in \mathbb{R}^{3 \times 4}$ (including both intrinsics and extrinsics), and per-view image masks $M^{1:V}$. For a *global* NeRF decoder, these are global non-background masks $M_{\text{tot}}^i \in \{0,1\}^{h \times w}$, and for a *compositional* NeRF decoder as in [53], these are sets of binary masks $M_j^i \in \{0,1\}^{h \times w}$ that identify the objects $j = 1, \ldots, m$ in the scene in view $i$. The global case is equivalent to $m = 1$, $M_{j=1}^i = M_{\text{tot}}^i$. The encoder $\Omega$ maps these posed image observations from the multiple views into a set of latent vectors $z_{1:m}$, where each $z_j$ represents each object in the scene separately in the compositional case, or the single $z_1$ all objects in the scene. This is achieved by querying $\Omega$ on the masks $M_j^{1:V}$, i.e.,

$$z_j = \Omega \left( I^{1:V}, K^{1:V}, M_j^{1:V} \right) \in \mathbb{R}^k \tag{2}$$

for object $j$. The supervision signal to train the encoder is the image reconstruction loss

$$\mathcal{L}^i = \left\| I^i \circ M_{\text{tot}}^i - D \left( \Omega \left( I^{1:V}, K^{1:V}, M_{1:m}^{1:V} \right), K^i \right) \right\|_2^2 \tag{3}$$

on the input view $i$ where the decoder $D$ renders an image $I = D(z_{1:m}, K)$ for arbitrary views specified by the camera matrix $K$ from the set of latent vectors $z_{1:m}$. Both the encoder and decoder

are trained end-to-end at the same time. The target images for the decoder are the same in both the global and compositional case: the global-masked image $I^i \circ M_{\text{tot}}^i$ ($\circ$ is the element-wise product). In the compositional case this can be computed with $M_{\text{tot}}^i = \bigvee_{j=1}^{m} M_j^i$. By fusing the information from multiple views of the objects into the latent vector from which the decoder has to be able to render the scene from multiple views, this auto-encoder framework can learn latent vectors that represent the 3D configurations (shape and pose) of the objects in the scene.

### 4.3 Latent-Conditioned NeRF Decoder Details

**Global.** The original NeRF formulation [16] learns a fully connected network $f$ that represents one single scene (Sec. 3.2). In order to create a decoder from NeRFs within an auto-encoder to learn a latent space, we condition the NeRF $f(\cdot, z)$ on the latent vector $z \in \mathbb{R}^k$ [42, 43, 44]. While approaches such as [42, 43, 44] use the latent code to represent factors such as lighting or category-level generalization, in our case the latent code is intended to represent the scene variation, i.e., shape *and* configuration of objects, such that a downstream RL agent may use this as a state representation.

**Compositional.** In the compositional case, the encoder produces a set of latent vectors $z_{1:m}$ describing each object $j = 1, \ldots, m$ individually, this leads to $m$ many NeRFs $(\sigma_j(x), c_j(x)) = f_j(x) = f(x, z_j)$, $j = 1, \ldots, m$ with their associated volume density $\sigma_j$ and color value $c_j$. Note that while one could use different networks $f_j$ with their own network weights for each object, we have a single network $f$ for all objects. This means that both the object's pose as well as its shape and type are represented through the latent code $z_j$. In order to force those conditioned NeRFs to learn the 3D configuration of each object separately, we compose them into a global NeRF model with the composition formulas (proposed e.g., by [80, 81]): $\sigma(x) = \sum_{j=1}^{m} \sigma_j(x)$, $c(x) = \frac{1}{\sigma(x)} \sum_{j=1}^{m} \sigma_j(x) c_j(x)$. As this composition happens in 3D space, the latent vectors will be learned such that they correctly represent the actual shape and pose of the objects in the scene with respect to the other objects, which we hypothesize may be useful for the downstream RL agent.

### 4.4 Encoder Details

The encoder $\Omega$ operates by fusing multiple views together to estimate the latent vector for the RL task. Since the scientific question of this work is to investigate whether a decoder built from NeRFs to train the encoder end-to-end is beneficial for RL, we consider two different encoder architectures. The first one is a 2D CNN that averages feature encodings from the different views, where each encoding is additionally conditioned on the camera matrix of that view. The second one is based on a learned 3D neural vector field that incorporates 3D biases by fusing the different camera views in 3D space through 3D convolutions and camera projection. This way, we are able to distinguish between the importance of 3D priors incorporated into the encoder versus the decoder.

**Per-image CNN Encoder ("Image encoder").** For the global version, we utilize the network architecture from [11] as an encoder choice. In order to work with multiple objects in the compositional case, we modify the architecture from [11] by taking the object masks into account as follows. For each object $j$, the 2D CNN encoder computes

$$z_j = \Omega_{\text{CNN}}\left(I^{1:V}, K^{1:V}, M_j^{1:V}\right) = h_{\text{MLP}}\left(\frac{1}{V}\sum_{i=1}^{V} g_{\text{MLP}}\left(E_{\text{CNN}}\left(I^i \circ M_j^i\right), K^i\right)\right). \qquad (4)$$

$E_{\text{CNN}}$ is a ResNet-18 [82] CNN feature extractor that determines a feature from the masked input image $I^i \circ M_j^i$ of object $j$ for each view $i$, which is then concatenated with the (flattened) camera matrix. The output of the network $g_{\text{MLP}}$ is hence the encoding of each view, including the camera information, which is averaged and then processed with $h_{\text{MLP}}$, to produce the final latent vector. Note that in the global case, we set $m = 1$, $M_{j=1}^i = M_{\text{tot}}^i$ such that $\Omega_{\text{CNN}}$ produces a single latent vector.

**Neural Field 3D CNN Encoder ("Field encoder").** Several authors [43] have considered to incorporate 3D biases into learning an encoder by computing pixel-aligned features from queried 3D locations of the scene to fuse the information from the different camera views directly in 3D space. We utilize the encoder architecture from [53], where the idea is to learn a neural vector field $\phi\left[I^{1:V}, M_j^{1:V}\right] : \mathbb{R}^3 \to \mathbb{R}^E$ over 3D space, conditioned on the input views and masks. The features of $\phi$ are computed from projecting the query point into the camera coordinate system from the respective view. To turn $\phi$ into a latent vector, it is queried on a workspace set $\mathcal{X}_h \in \mathbb{R}^{d_{\mathcal{X}} \times h_{\mathcal{X}} \times w_{\mathcal{X}}}$ (a 3D

grid) and then processed by a 3D convolutional network, i.e., $z_j = E_{\text{3D CNN}}\left(\phi\left[I^{1:V}, M_j^{1:V}\right](\mathcal{X}_h)\right)$. This method differs from [43, 83, 60] by computing a latent vector from the pixel-aligned features.

## 5    Baselines / Alternative State Representations

In this section, we briefly describe alternative ways of training an encoder for RL, which we will investigate in the experiments as baselines and ablations. For details, refer to the appendix.

**Conv. Autoencoder.** This baseline uses a standard CNN decoder based on deconvolutions instead of NeRF to reconstruct the image from the latent representation, similar to [1]. Therefore, with this baseline we investigate the influence of the NeRF decoder relative to CNN decoders. We follow the architecture of [11] for the deconvolution part for the global case. In the compositional case, we modify the architecture to be able to deal with a set of individual latent vectors instead of a single, global one. The image $I = D_{\text{deconv}}(g_{\text{MLP}}(\frac{1}{m}\sum_{j=1}^{m} z_j), K)$ is rendered from $z_{1:m}$ by first averaging the latent vectors and then processing the averaged vector with a fully connected network $g_{\text{MLP}}$, leading to an aggregated feature. This aggregated feature is concatenated with the (flattened) camera matrix $K$ describing the desired view and then rendered into the image with $D_{\text{deconv}}$. In the experiments, we utilize this decoder as the supervision signal to train the latent space produced by the 2D CNN encoder from Sec. 4.4. In the compositional version, the 2D CNN encoder (4) use the same object masks as the compositional NeRF-RL variant.

**Contrastive Learning.** As an alternative to learning an encoder via a reconstruction loss, the idea of contrastive learning [84] is to define a loss function directly on the latent space that tries to pull latent vectors describing the same configurations together (called positive samples) while ones representing different system states apart (called negative samples). A popular approach to achieve this is with the InfoNCE loss [85, 64]. Let $y_i$ and $\tilde{y}_i$ be two *different* observations of the *same* state. Here, $\tilde{\cdot}$ denotes a perturbed/augmented version of the observation. For a mini-batch of observations $\{(y_i, \tilde{y}_i)\}_{i=1}^n$, after encoding those into their respective latent vectors $z_i = \Omega(y_i)$, $\tilde{z}_i = \Omega(\tilde{y}_i)$ with the encoder $\Omega$, the loss for that batch would use $(z_i, \tilde{z}_i)$ as a positive pair, and $(z_i, \tilde{z}_{\neq i})$ as a negative pair, or some similar variation. A crucial question in contrastive learning is how the observation $y$ is perturbed/augmented into $\tilde{y}$ to generate positive and negative training pairs, described in the following.

**CURL.** In CURL [5], the input image is randomly cropped to generate $y$ and $\tilde{y}$. We closely follow the hyperparameters and design of [5]. CURL operates on a single input view and we choose a view for this baseline from which the state of the environment can be inferred as best as possible (Fig. 17).

**Multi-View CURL.** This baseline investigates if the neural field 3D encoder (Sec. 4.4) can be trained with a contrastive loss. As this encoder operates on multiple input views we *double* the number of available camera views. Half of the views are the same as in the other experiments, the other half are captured from sightly perturbed camera angles. We use the same loss as CURL, but with different contrastive pairs – rather than from augmentation, the contrastive style is taken from TCN [68]: the positive pairs come from different views but at the same moment in time, while negative pairs come from different times. Therefore, this baseline can be seen as a multi-view adaptation of CURL [5].

**Direct State / Keypoint Representations.** Finally, we also consider a direct, low-dimensional representation of the state. Since we are interested in generalizing over different object shapes, we consider multiple 3D keypoints that are attached at relevant locations of the objects by expert knowledge and observed with a perfect keypoint detector [8]. See Fig. 2b for a visualization of those keypoints. The keypoints both provide information about object shape and its pose. Furthermore, as seen in Fig. 2b, they have been chosen to reflect those locations in the environment relevant to solve the task. Additionally, we report results where the state is represented by the poses of the objects – as this cannot represent object shape, in this case we use a constant object shape for training and test.

## 6    Experiments

We evaluate our proposed method on different environments where the geometry of the objects in the scene is important to solve the task successfully. Please also refer to the video https://dannydriess.github.io/nerf-rl. Commonly, RL is trained and evaluated on a single environment, where only the poses are changed, but the involved object shapes are kept constant. Since latent-conditioned NeRFs have been shown to be capable of generalizing over geometry [43],

we consider experiments where we require the RL agent to generalize over object shapes within some distribution. Answering the scientific question of this work requires environments with multi-view observations — and for the compositional versions object masks as well. These are *not provided in standard RL benchmarks*, which is the reason for choosing the environments investigated in this work. We use PPO [86] as the RL algorithm and four camera views in all experiments. Refer to the appendix for more details about our environments, parameter choices, network architectures, and training times.

## 6.1   Environments

**Mug on Hook.**   In this environment, adopted from [87] and visualized in Fig. 2b, the task is to hang a mug on a hook. Both the mug and the hook shape are randomized. The actions are small 3D translations applied to the mug. This environment is challenging as we require the RL agent to generalize over mug and hook shapes and the tolerance between the handle opening and the hook is relatively small. Further, the agent receives a sparse reward only if the mug has been hung stably. This reward is calculated by virtually simulating a mug drop after each action. If the mug does not fall onto the ground from the current state, a reward of one is assigned, otherwise zero.

**Planar Pushing.** The task in this environment, shown in Fig. 3b, is to push yellow box-shaped objects into the left region of the table and blue objects into the right region with the red pusher that can move in the plane, i.e., the action is two dimensional. This is the same environment as in [53] with the same four different camera views. Each run contains a single object on the table (plus the pusher). If the box has been pushed inside its respective region, a sparse reward of one is received, otherwise zero. The boxes in the environment have different sizes, two colors and are randomly initialized. In this environment, we cannot use keypoints for the multi-shape setting, as the reward depends on the object color; we evaluate the keypoints baseline only in the single shape case (Appendix).

**Door Opening.** Fig. 4b shows the door environment, where the task is to open a sliding door with the red end-effector that can be translated in 3 DoFs as the action. To solve this task, the agent has to push on the door handle. As the handle position and size is randomized, the agent has to learn to interact with the handle geometry accordingly. Interestingly, as can be seen in the video in the supplementary material, the agent often chooses to push on the handle only at the beginning, as, afterwards, it is sufficient to push the door itself at its side. The agent receives a sparse reward if the door has been opened sufficiently, otherwise, zero reward is assigned.

## 6.2   Results

Figs 2a, 3a, 4a show success rates (averaged over 6 independent experiment repetitions and over 30 test rollouts per repetition per timestep) as a function of training steps. Also shown are the $68\%$ confidence intervals. These success rates have been evaluated using randomized object shapes and initial conditions, and therefore reflect the agent's ability to generalize over these.

In all these experiments, a latent space trained with compositional NeRF supervision as the decoder consistently outperformed all other learned representations, both in terms of sample efficiency and asymptotic performance. Furthermore, our proposed framework with compositional NeRF even outperforms the expert keypoint representation. For the door environment, the 3D neural field encoder plus NeRF decoder (NeRF-RL comp. + field) reaches nearly perfect success rates. For the other two environments, the compositional 2D CNN encoder plus NeRF decoder (NeRF-RL comp. + image) was slightly better than with the neural field encoder but not significantly. This shows that the *decoder* built from compositional NeRF is relevant for the performance, not so much the choice of the encoder.

Training the 3D neural field encoder with a contrastive loss as supervision signal for different camera views as positive/negative training pairs is not able to achieve significant learning progress in these scenarios (Multi-CURL). However, the other contrastive baseline, CURL, which has a different encoder and uses image cropping as data augmentation instead of additional camera views, is able to achieve decent performance and sample efficiency on the door environment, but not for the pushing environment. In the mug environment, CURL initially is able to make learning progress comparable to our framework, but never reaches a success rate above 59% and then becomes unstable. Similarly, the global CNN autoencoder baseline shows decent learning progress initially on the mug and pushing scenario (not for the door), but then becomes unstable (mug) or never surpasses 50% success rate

|  |  | encoder | decoder | comp. | NeRF | loss |
|---|---|---|---|---|---|---|
| NeRF-RL (ours) | **comp.+field** | 3D CNN | comp. 3D NeRF | ✓ | ✓ | image reconstr.: L2 |
|  | **comp.+image** | 2D CNN | comp. 3D NeRF | ✓ | ✓ | image reconstr.: L2 |
|  | **global+image** | 2D CNN | global 3D NeRF | ✗ | ✓ | image reconstr.: L2 |
|  | Conv. Autoencoder, $c$ | 2D CNN | comp. 2D CNN | ✓ | ✗ | image reconstr.: L2 |
|  | Conv. Autoencoder, $g$ | 2D CNN | 2D CNN | ✗ | ✗ | image reconstr.: L2 |
|  | CURL | 2D CNN | - | ✗ | ✗ | contrast: InfoNCE |
|  | Multi-CURL | 3D CNN | - | ✓ | ✗ | contrast: InfoNCE |
|  | Keypoints | chosen by expert knowledge and perfect extraction | | | | |

Table 1: Overview of the different state representation learning frameworks.

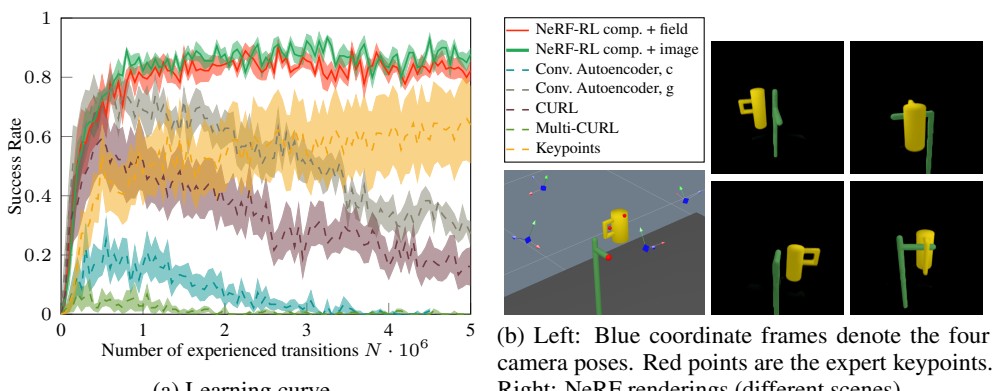

(a) Learning curve.

(b) Left: Blue coordinate frames denote the four camera poses. Red points are the expert keypoints. Right: NeRF renderings (different scenes).

Figure 2: Mug on hook environment. (b) shows an example scene and NeRF renderings

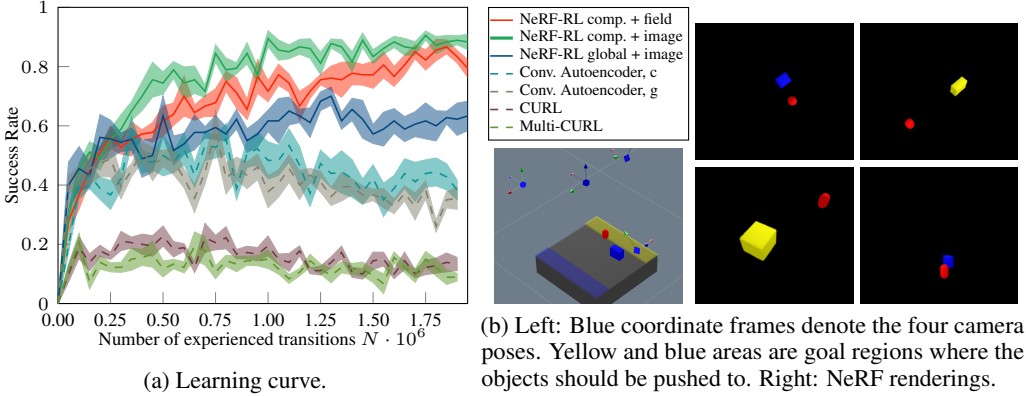

(a) Learning curve.

(b) Left: Blue coordinate frames denote the four camera poses. Yellow and blue areas are goal regions where the objects should be pushed to. Right: NeRF renderings.

Figure 3: Pushing environment. (b) shows NeRF renderings for different scenes.

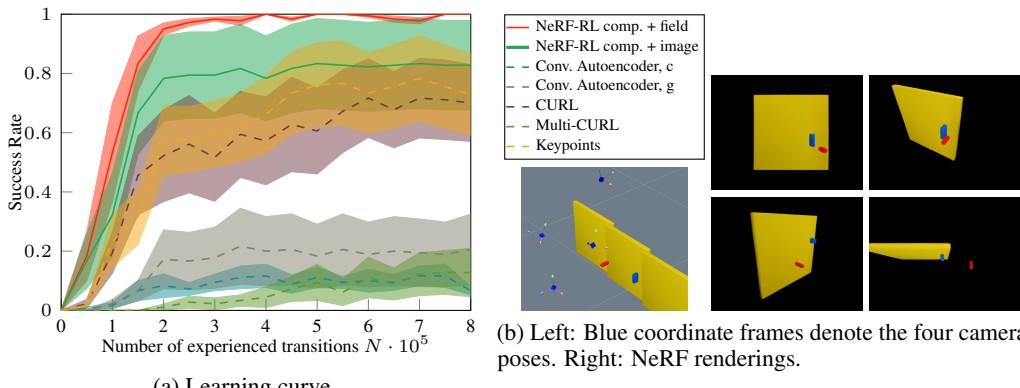

(a) Learning curve.

(b) Left: Blue coordinate frames denote the four camera poses. Right: NeRF renderings.

Figure 4: Door environment. (b) shows NeRF renderings for different scenes.

(pushing). Such variations in performance or instable learning across the different environments have not been observed with our method, which is stable in all cases.

The compositional variant (NeRF-RL comp.) of our framework achieves the highest performance. Since the conv. comp. autoencoder baseline has worse performance than its global variant, compositionality alone is not the sole reason for the better performance of our state representation. Indeed, the global NeRF-RL + image variant in the pushing env. is also better than all other baselines.

In the appendix Sec. A.1, we find a positive correlation between NeRF reconstruction quality and RL performance. Furthermore, it turns out that the performance of our framework is not significantly affected when we pretrain the encoder with less data (Sec. A.2). In Sec. A.3, we investigate the influence of the number of input views on the RL performance. In the pushing scenario, only two or even one input view are sufficient for good performance. However, for tasks that require more 3D understanding such as the mug scenario, we observe a drop in performance when reducing the number of views from 4 to 2.

## 7   Discussion

**Why NeRF provides better supervision.**   The NeRF training objective (1) strongly forces each $f(\cdot, z_j)$ to represent each object in its actual 3D configuration and relative to other objects in the scene (compositional case), including their shape. This implies that the latent vectors $z_j$ have to contain this information, i.e., they are trained to determine the object type, shape and pose in the scene. In the global case, $z_1$ has to represent the geometry of the whole secne. As the tasks we consider require policies to take the geometry of the objects into account, we hypothesize that a latent vector that is capable of parameterizing a NeRF to reconstruct the scene in the 3D space has to contain enough of the relevant 3D information of the objects also for the policy to be successful.

**Masks.**   In order for the auto-encoder framework to be compositional, it requires object masks. We believe that instance segmentation has reached a level of maturity [88] that this is a fair assumption to make. As we also utilize the individual masks for the compositional conv. autoencoder and the multi-view CURL baseline, which do not show good performance, it indicates that the masks are not the main reason that our state representation achieves higher performance. This is further supported by the fact that the global NeRF-RL variant which does not rely on individual object masks on the pushing scenario achieved a performance higher than all baselines, i.e., masks will increase the performance of NeRF-RL as they enable the compositional version, but they do not seem essential.

**Offline/Online.**   In this work, we focused on pretraining the latent representation offline from a dataset collected by random actions. During RL, the encoder is fixed and only the policy networks are learned. This has the advantage that the same representation can be used for different RL tasks and the dataset to train the representation not necessarily has to come from the same distribution. However, if a policy is needed to explore reasonable regions of the state space, collecting a dataset offline to learn a latent space that covers the state space sufficiently might be more challenging for an offline approach. This was not an issue for our experiments where data collection with random actions was sufficient. Indeed, we show generalization over different starting states of the same environment and with respect to different shapes (within distribution). Future work could investigate NeRF supervision in an online setup. Note that the reconstruction loss via NeRF is computationally more demanding than via a 2D CNN deconv. decoder or a contrastive term, making NeRF supervision as an auxiliary loss at each RL training step costly. One potential solution for this is to apply the auxiliary loss not at every RL training step, but with a lower frequency. Regarding computational efficiency, this is where contrastive learning has an advantage over our proposed NeRF-based decoder, as the encoding with CURL can be trained within half a day, whereas the NeRF auto-encoder took up to 2 days to train for our environments. However, when using the encoder for RL, there is no difference in inference time.

**Multi-View.**   The auto-encoder framework we propose can fuse the information of multiple camera views into a latent vector describing an object in the scene. This way, occlusions can be addressed and the agent can gain a better 3D understanding of the scene from the different camera angles. Having access to multiple camera views and their camera matrices is an additional assumption we make, although we believe the capability to utilize this information is an advantage of our method.

# 8 Conclusion

In this work, we have proposed the idea to utilize Neural Radiance Fields (NeRFs) to train latent spaces for RL. Our environments focus on tasks where the geometry of the objects in the scene is relevant for successfully solving the tasks. Training RL agents with the pretrained encoder that maps multiple views of the scene to a latent space consistently outperformed other ways of learning a state representation and even keypoints chosen by expert knowledge. Our results show that the 3D prior present in compositional NeRF as the decoder is more important than priors in the encoder.

**Broader Impacts.** Our main contribution is a method to learn representations that improve the efficiency of vision-based RL, which could impact automation. As such, our work inherits general ethical risks of AI, like the question of how to address the potential of increased automation in society.

## Acknowledgments

The authors thank Russ Tedrake for initial discussions; Jonathan Tompson and Jon Barron for feedback on drafts; Vincent Vanhoucke for encouraging latent NeRFs.

This research has been supported by the Deutsche Forschungsgemeinschaft (DFG, German Research Foundation) under Germany's Excellence Strategy – EXC 2002/1 "Science of Intelligence" – project number 390523135. Danny Driess thanks the International Max-Planck Research School for Intelligent Systems (IMPRS-IS) for the support. Ingmar Schubert acknowledges support by the German Academic Scholarship Foundation. Yunzhu Li acknowledges support by Amazon.com Services LLC, PO# #2D-06310236 and the Wistron Corporation.

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
