# OpenReview forum: "Reinforcement Learning with Neural Radiance Fields"
_NeurIPS.cc/2022/Conference — NeurIPS 2022 Accept_

### Official Review · Reviewer_aWEL · 2022-07-06

**Rating:** 5
**Confidence:** 5
**Soundness:** 3 good
**Presentation:** 3 good
**Contribution:** 2 fair

**Summary:**

The paper propose a Neural Radiance Fields-based RL (NeRF-RL) method, which makes use of supervision from NeRF to learn state representations describing the objects in the 3D scene. Several experiments involving robotic object manipulations were conducted to demonstrate the superior performance of NeRF-RL over existing methods.

**Questions:**

Q1. Does NeRF-RL limit to 3D scenes? Can NeRF-RL be used to solve 2D RL tasks such as Atari games? I think it is necessary to discuss whether and how NeRF-RL applies to general RL tasks.

Q2. NeRF-RL requires plenty of offline RL data to train the NeRF encoder before the policy training process. However, offline RL data are difficult to obtain for realistic 3D environments.

Q3. As mentioned before, all the experimental scenes are simple and clear. Can  NeRF-RL generalize well to complicated 3D scenes with a large number of objects?

Q4. Does NeRF require more offline RL data and larger neural network to learn state representations than baseline methods?

**Limitations:**

See the questions.

**Strengths And Weaknesses:**

###########Strengths###########

The paper is well written, and motivated. I believe that the experiments are roughly solid and reproducible.

###########Weaknesses###########

The paper is lack of adequate analysis and discussion about the limitation of NeRF-RL. The experimental scenes are simple and clear, and have only a small number of objects. I concern about whether NeRF-RL can generalize well to complicated 3D scenes, such as VizDoom and Minecraft.

---

> ### Author Response · Authors · 2022-08-02
> **Thank you for your review! (1)**
>
> >The paper is lack of adequate analysis and discussion about the limitation of NeRF-RL. The experimental scenes are simple and clear, and have only a small number of objects.
>
> Thanks a lot for your review, which contains valuable questions that we try to address with additional experiments. In particular, we have added experiments regarding data-efficiency, a 2D task, and a scenario containing more objects with which we hope to shine more light on the limitations of NeRF-RL. We will describe these experiments in more detail below, answering your questions directly.
>
> >I concern about whether NeRF-RL can generalize well to complicated 3D scenes, such as VizDoom and Minecraft.
>
> Generalizing to very complex or real-world scenes will require NeRF encoders that can successfully represent such environments. Encouragingly, there is already great progress in this area, for example Block-NeRF (M Tancik et al., 2022). We don't investigate such scenarios here, but we believe this would be an interesting (and promising!) question for future work.
>
> >Does NeRF-RL limit to 3D scenes? Can NeRF-RL be used to solve 2D RL tasks such as Atari games? I think it is necessary to discuss whether and how NeRF-RL applies to general RL tasks
>
> Thank you! We agree that this is an important point. We added an additional experiment to investigate the applicability of our method to a 2D scenario (see below). Since 2D scenarios can be seen as special cases of 3D scenarios, we think (and the experiment suggests) that our method can also be used to solve 2D RL tasks. However, our method is particularly suitable for tasks requiring a good 3D understanding of the environment.
>
> Nevertheless, in order to show that NeRF-RL is not necessarily limited to 3D scenarios, we revisited the pushing scenario and trained the NeRF state representation on a single, top-down view from above the table. Since in this environment, all movement is on the table, and since the view is top-down, this is a 2D task. As shown in the appendix Sec. A.3, NeRF-RL still outperforms all other baselines in this setup.
>
> We added these experiments along with a more detailed discussion of the applicability of our method to the revised version of the paper (Sec. A.3).
>
> >NeRF-RL requires plenty of offline RL data to train the NeRF encoder before the policy training process. However, offline RL data are difficult to obtain for realistic 3D environments.
>
> Thank you! We have added an experiment in Sec. A.2 where we significantly reduce the amount of offline RL training data. In the box pushing environment, we find that using 1/10th of the data does not decrease performance of the RL agent. Likewise, preliminary results for the mug hanging environment suggest the same for half the data (the runs are not fully finished yet, but the learning curve has converged, and will be updated again in the final version).
>
> Independently from these results, we would like to point out that the amount of offline RL data used to train the state representation is orders of magnitudes smaller than during online RL. Furthermore, the offline RL data does not cover the possible scene space densely. For instance, the mug scenes are sampled from an 11 dimensional (3D position + diameter + height + handle position + handle height + handle extend + hook height + hook vertical position + hook length) parameter space, while only 10,000 (or 5,000 for the new experiment) data points have been used to train the state representation. This only amounts to an average density of 10,000^(1/11)=2.3 samples per dimension.
>
> >As mentioned before, all the experimental scenes are simple and clear. Can NeRF-RL generalize well to complicated 3D scenes with a large number of objects?
>
> Thank you! To address this, we have added an experiment (appendix Sec. A.4) where the trained policy of the box pushing scenario is applied to scenes that contain more than one object. No retraining was necessary to achieve these results. We find that NeRF-RL outperforms the baselines in this scenario as well. We also updated the supplementary material with example videos of this experiment.
>
> As mentioned above, generalizing to very complex or real-world scenes will require NeRF encoders that can successfully represent such environments, for which methods such as Block-NeRF (M Tancik et al., 2022) show promising results.

---

> > ### Author Response · Authors · 2022-08-02
> > **Thank you for your review! (2)**
> >
> > >Does NeRF require more offline RL data and larger neural network to learn state representations than baseline methods?
> >
> > Thank you! NeRF-RL does not require more offline RL data compared to the baseline methods; all baselines used the same amount of training data as NeRF-RL. Furthermore, as mentioned above and shown in the appendix Sec. A.2, NeRF-RL still outperforms the other methods when the encoder is trained on much less data.
> >
> > All baselines (except CURL, see below) utilize the same network sizes where applicable. For CURL, we used the same architecture as proposed in the original publication, which resulted in NeRF-RL having larger network sizes for the mug and door environment, but a smaller network size in the box scenario than the CURL baseline. We believe that none of our findings indicate that the network size played a role here.
> >
> > We added a brief discussion of both these aspects to Sec. B of the updated version of the paper.

---

> > > ### Comment · Reviewer_aWEL · 2022-08-05
> > > **Response**
> > >
> > > Thanks for the response that addresses a part of my concerns. I am not sure whether the contribution is sufficient to publish in NeurIPS, thus mantain the score unchanged. If the authors generalize the method to more complex 3D environments, I will raise the score to at least 6.

---

> > > > ### Author Response · Authors · 2022-08-07
> > > > **Response**
> > > >
> > > > Thank you for your answer and thank you for acknowledging that we addressed part of your concerns.
> > > >
> > > > We would like to point out that our environments are more challenging than they might seem. In contrast to most RL tasks (including VizDoom), our environments have a distribution of object shapes (instead of fixed object shapes). In order to be successful, the RL agent has to take the (fine) geometry of the objects into account. The tasks require precise continuous actions, as, e.g., the margin in the mug scenario between the hook and the mug handle is relatively small. Our method is compositional, i.e. it supports generalization to scenes containing more objects, which we have shown in response to your comments.
> > > >
> > > > As acknowledged by the other reviewers, our work is the first paper (that we are aware of) to propose to incorporate NeRF in RL, which we have shown has benefits compared to multiple baselines, and therefore, in our opinion, is a valuable contribution to the community.

---

### Official Review · Reviewer_WpRR · 2022-07-11

**Rating:** 7
**Confidence:** 4
**Soundness:** 4 excellent
**Presentation:** 3 good
**Contribution:** 4 excellent

**Summary:**

The authors proposed the first learned state representation through NeRF supervision for reinforcement learning purposes.
The proposed method shows that an offline trained latent-conditioned NeRF comes with a learned state representation that can be beneficial for training an RL policy online more sample efficiently.
The final results on three simulated environments are impressive compared to other learned representations and a key-points representation selected by human experts.

**Questions:**

* Any thoughts on how the proposed method can be generalized to real-world environments?


**Limitations:**

I didn't find a clear section discussing the limitation.

**Strengths And Weaknesses:**

Pros:
* the first learned state representation through NeRF supervision for reinforcement learning purposes.
* the sample efficiency has greatly improved compared to other learned representations and a key-points representation selected by human experts.

Cons:
* Only simplified simulation environments are used rather than real-world environments rendered by NeRF.

---

> ### Author Response · Authors · 2022-08-02
> **Thank you for your review!**
>
> >Any thoughts on how the proposed method can be generalized to real-world environments?
>
> Thank you! There is indeed an existing body of work on training NeRF on real-world scenes, such as NeRF in the Wild (R Martin-Brualla et al., 2021) or Block-NeRF (M Tancik et al., 2022). These works show that it is feasible to learn NeRF representations even for complex real-world scenarios. In the present work, our aim was to focus on investigating the research question whether latent representations learned with NeRF supervision can benefit RL. Hence, we think it is important to compare our method to a wide variety of baselines, and ablate/test our method in different scenarios, for which simulated environments offer a more suitable testbed.
>
> Although we did not consider real-world scenarios in this work, given the above mentioned advances in NeRF on real world scenarios, we don’t see a priori reasons why our proposed method should not generalize to such real-world representations as well.

---

### Official Review · Reviewer_iBq4 · 2022-07-11

**Rating:** 6
**Confidence:** 4
**Soundness:** 3 good
**Presentation:** 3 good
**Contribution:** 3 good

**Summary:**

The paper presents a method that uses a Neural Radiance Field representation as a state space for a reinforcement learning algorithm.  This approach is evaluated in tasks where multiple views of the scene are available, and where the 3D information present in the NeRF representation is useful for determining shape and pose of objects.  The paper explores settings where the NeRF represents the scene globally, and also where a compositional architecture allows representing each object with its own radiance field, and shows improved performance over several baselines.

## update after author response

Thank you to the authors for a diligent and thorough response. I have raised my score.

**Questions:**

- I would have liked to see more discussion of the sensitivity of the RL performance to, say, the data available to train the NeRF model.  What sort of coverage of the scene by input views is required?
- I am surprised at the poor performance of the keypoint method.  I would guess that appropriately chosen 3D keypoint data should represent an upper bound of possible performance for the NeRF method.  What sort of limitation does the keypoint method have here that the NeRF method outperforms it?  In the door case, for example, what information besides the keypoints would the NeRF represent?

**Limitations:**

The authors are responsible about pointing out the assumptions they make and limitations of the presented approach, and discuss these assumptions and limitations in an up-front way.

**Strengths And Weaknesses:**

### Originality
The major contribution of the paper is using NeRFs as a representation specifically in RL, and although this is not a surprising combination of methods, it is the first paper I am aware of that uses RL on top of the NeRF representation.  The paper cites many similar approaches which use NeRFs in dynamics models which are then used for planning (though I believe the authors may want to additionally cite [Li, Li, Sitzmann, Agrawal, and Torralba. 3D Neural Scene Representations for Visuomotor Control. CoRL 2021]).  The substitution of RL for planning is a small contribution.

### Quality
The paper does well to address different methods for using NeRFs as a representation, both global and compositional, and has a thorough list of baselines for comparison.  The focus is on highlighting the advantages of using the 3D aware representations that come from a NeRF.  However, the tasks are specifically designed to increase the likelihood that this 3D information is useful, and discussion of how sensitive the approach is to those assumptions is not addressed.

### Clarity
The paper is well written and the method is described with enough background to give a self-contained understanding of the approach.  There are some issues and minor typos:
- Line 139 supevision -> supervision
- Line 188 then -> them
- In description of the CURL baseline, the text mentions carefully choosing views, but it is not clear how this is done.

### Significance
This paper presents a small but interesting contribution.  As the authors mention, it is unclear what realistic situations will actually provide multiple viewpoints of the type given in the simulations here, but this is a first glimpse to suggest that NeRFs can help.

---

> ### Author Response · Authors · 2022-08-02
> **Thank you for your review! (1)**
>
> >The paper cites many similar approaches which use NeRFs in dynamics models which are then used for planning (though I believe the authors may want to additionally cite [Li, Li, Sitzmann, Agrawal, and Torralba. 3D Neural Scene Representations for Visuomotor Control. CoRL 2021]).
>
> Thank you for pointing out this work! We actually cited it in the original version already (reference 11) and agree that it is relevant to our approach.
>
> >The substitution of RL for planning is a small contribution.
>
> We agree that the high-level idea is simple, but, as you acknowledged, using NeRF-representations for RL has not been proposed before. We believe that it is a scientifically interesting question whether learning state representations with NeRF supervision is not only advantageous for learning dynamic models as in previous work, but also for learning policies/Q-functions and RL in general.  Also, while the prior work in dynamics models showed benefitting from more complicated model design (i.e. graph neural nets on top of compositional latents), here we show that for model-free RL a simple design of just feeding in the NeRF latents to a standard RL model can be quite successful.
>
> >However, the tasks are specifically designed to increase the likelihood that this 3D information is useful, and discussion of how sensitive the approach is to those assumptions is not addressed.
>
> Thanks for raising this point which we try to address with an additional experiment (see below). We agree that we have indeed focused on 3D tasks. This is because we believe that 3D tasks in general are more challenging, and that there are fewer methods developed that are suitable for 3D tasks where the geometry of the objects is important.
>
> Having said that, in order to show that NeRF-RL is not necessarily limited to 3D scenarios, we revisited the pushing scenario and trained the NeRF state representation on a single, top-down view from above the table. Since almost all movement happens on the table, this setup results in a 2D task. We present the results in the appendix Sec. A.3. It turns out that, in this setup, even with this top-down view only, both versions of NeRF-RL achieve a similar performance as their counterparts using 4 views (the learning curves are statistically indistinguishable). Therefore, NeRF-RL still outperforms the baselines in this case, even if it only has access (and is only trained with) a single 2D top-down view of the scene.
>
> >There are some issues and minor typos: [...]
>
> Thank you, we have fixed these in the revised version.
>
> >In description of the CURL baseline, the text mentions carefully choosing views, but it is not clear how this is done.
>
> Thank you! We updated the revised version of the paper with a figure showing these views (Fig. 17):
>
> Especially for the mug and door task, if a view is chosen for the single-view CURL baseline that occludes crucial parts of the scene (like the position of the handle), the resulting representation will not contain enough information about the scene for the policy to succeed. Therefore, we manually selected a view for the CURL baseline that enables the encoder to infer the state of the system as best as possible.

---

> > ### Author Response · Authors · 2022-08-02
> > **Thank you for your review! (2)**
> >
> > >I would have liked to see more discussion of the sensitivity of the RL performance to, say, the data available to train the NeRF model.
> >
> > Thank you for raising this question. We agree that this is interesting and deserves a more in-depth discussion. We have added multiple experiments that investigate the sensitivity of the RL performance to the data available to train the NeRF model. In particular, we trained the NeRF encoder with 1/2 of the data in the mug hanging environment, and with 1/10 of the data in the box pushing environment. The results of these additional experiments are reported in the appendix Sec. A.2. The runs for the mug environment are still ongoing, but the learning curve has converged. We will update the results in the final version with the full results of the completed runs.
> >
> > In the box environment, it turns out that using considerably smaller amounts of data (1/10th) does not decrease performance of the RL agent. Likewise, we find the same if we reduce the data in the mug environment to 1/2. Additionally, we would like to point out that, for example, in the mug hanging environment, the data coverage to train the NeRF model was quite sparse to begin with. The configuration space of the mug environment (3D position + diameter + height + handle position + handle height + handle extend + hook height + hook vertical position + hook length) is 11-dimensional. For the experiments in the original paper, we used 10000 data points, which corresponds to an average density of 10000^(1/11)=2.3 points per dimension. Even though the data coverage to train the encoder is quite sparse, NeRF-RL is still able to effectively interpolate to other shapes. Please refer to the appendix Sec. B.1 for a discussion of this.
> >
> > Additionally, as discussed, e.g., in Sec. B.3, the trajectories in the offline dataset are quite different to those during online RL.
> >
> > >What sort of coverage of the scene by input views is required?
> >
> > Thank you for raising this interesting point. Following your remark, we investigated experimentally whether our method works with only two or even only one viewpoint, and find that it is quite robust in these scenarios (see below for details). Having said that, we would argue that many relevant applications in robotics will provide multiple views, e.g. by mounting multiple cameras or having a camera mounted to a moving part of the robot. In these situations, we believe that it is an advantage of our method to be able to fuse those multiple camera views.
> >
> > In the pushing environment, we find that both for NeRF-RL comp. + field and NeRF-RL comp. + image, using less viewpoints does not decrease performance significantly. This can be explained by the fact that the pushing task is mostly 2D. We repeated the same experiment for the mug environment, which requires more detailed 3D information. Here, our runs are not fully finished yet, but the learning curve has converged to a point where they already suggest that using less viewpoints decreases the performance to some extent, but still outperforms multiple baselines having access to all views. In the revised version of the paper, we report these additional experiments in Sec. A.3.

---

> > > ### Author Response · Authors · 2022-08-02
> > > **Thank you for your review! (3)**
> > >
> > > >I am surprised at the poor performance of the keypoint method. I would guess that appropriately chosen 3D keypoint data should represent an upper bound of possible performance for the NeRF method. What sort of limitation does the keypoint method have here that the NeRF method outperforms it? In the door case, for example, what information besides the keypoints would the NeRF represent?
> > >
> > > Thank you. We agree that carefully chosen keypoints are an important and interesting baseline. Following your comment, we added a more detailed experiment on this (see below for details). Having said that, we would like to argue that keypoints do not necessarily represent an upper bound of possible performance, for the following reason:
> > >
> > > All of the state representations we use throughout in theory could either be learned such, or, in case of the keypoints, are defined such that they contain the full state information of the MDP. The crucial difference is in how this information is represented. Some representations allow for a more efficient training of the neural network than others. We would argue that it is not clear a priori which representation is the most efficient one. According to our experiments, it is the case that the NeRF representation facilitates faster learning than the keypoint representation. In that sense, the limitation of the keypoint method is not that it contains less information, but rather that the information is encoded in such a way that “makes it harder” for the neural network to pick up on it.
> > >
> > > We agree that the paper would benefit from a more careful discussion of the choice of keypoints. We therefore added an additional experiment in the mug environment. This time, we choose a larger number of keypoints, with additional ones located at the corner points of the mug handle, and on the opposite side of the mug (Fig. 11). Results show that the performance with more keypoints is in fact slightly better than the keypoints baseline shown in the original paper, but is still surpassed by compositional NeRF-RL. This suggests that keypoints are a less efficient representation of the state in this case, regardless of how they are chosen. We updated the paper with the additional learning curve (Fig. 10a) and discussion in Sec. A.5.

---

> > > > ### Comment · Reviewer_iBq4 · 2022-08-07
> > > > **Keypoints**
> > > >
> > > > Thank you for addressing this concern with the additional experiment.  It is indeed interesting that the increased number of keypoints improves performance.  I understand your arguments about representations which facilitate learning, I am just surprised that the keypoint representation in these simple tasks is hard to learn with.  Is there some other side-effect of the keypoint representation that may be causing this difference?  Do, for example, the keypoint-based agents have some different 'capacity' of their architecture since the input state representation has a different dimensionality?

---

> > > > > ### Author Response · Authors · 2022-08-09
> > > > > **Thank you! We added more experiments on the keypoint baseline**
> > > > >
> > > > > Thanks for getting back to us! We agree that the questions you raise are interesting. Therefore, following your remarks, we experimentally investigated the keypoints baseline further. We added the following experiments and report them in Sec. A.5 and Fig. 12 of the updated version of the paper:
> > > > > 1. For the keypoints baseline, we increased the size of the first hidden layer of the value and policy networks in such a way that the total number of parameters of the policy and value networks is the same as for NeRF-RL (due to the smaller input dimension of the keypoint baseline, the agent networks indeed had a smaller number of parameters originally). Note that all other baselines using latent state representations reported in the paper already had the exact same agent network sizes.
> > > > > 2. We used even more keypoints, making the dimension of the keypoint representation equal to the size of the NeRF representation. Hence, also in this case, there is the same amount of parameters of the agent networks as in NeRF-RL
> > > > >
> > > > > For both (1) and (2), we find that with your suggestion we can further improve upon both keypoints baselines that we discussed in the original paper and in our first answer to your review. Still, NeRF-RL outperforms all four keypoint baselines, especially in terms of learning speed. The final asymptotic performance of the improved keypoint baseline (1) is comparable to NeRF-RL.
> > > > >
> > > > > We would like to mention that this keypoint baseline not only uses expert knowledge of the environment, but also is now quite tuned. No tuning of network hyperparameters was necessary for achieving the results for NeRF-RL. Both variants of NeRF-RL (image + field) have similar performance, while out of the 4 keypoints baselines, only one was able to reach asymptotic performance similar to NeRF-RL.
> > > > >
> > > > > > the keypoint representation [...] is hard to learn with
> > > > >
> > > > > We would not nevessarily say that the keypoint representation baseline performs badly, especially after the modifications in response to your comments; it is just the case that NeRF-RL is better in terms of learning speed and did not require as much tuning.

---

> ### Author Response · Authors · 2022-08-07
> **Reviewer-author discussion period ends on Tuesday**
>
> Thank you again for your review. We hope that we were able to address your concerns in our response. If possible, we would appreciate a response before the reviewer-author discussion period ends.

---

### Author Response · Authors · 2022-08-02
**Authors Initial Response - Overview**

We would like to thank the reviewers for their constructive and thorough feedback. We tried to address their concerns and revised the paper accordingly, and will also answer directly to their reviews. The following is an overview of all updates/additions to the paper (marked in blue in the PDF). We also updated the supplementary material with an additional video.

- We added new experiments investigating the performance of NeRF-RL if less data is used to train the NeRF model. NeRF still accomplishes the task with no significant drop in performance with just 1/2 (mug environment) or 1/10 (pushing environment) of training data. [Answer to iBq4, aWEL]
- We added an experiment in the table-top pushing environment using only one top-down view of the scene, thus creating a 2D setup, as the movement of the objects happens on the table as well. We find that our method’s performance is not significantly affected in this case. [Answer to iBq4, aWEL]
- We added experiments investigating the performance of NeRF-RL if the NeRF model is trained on fewer views per scene. In the mug scenario, we observe that the performance is decreased as we decrease the number of views, which is expected since 3D information is important here. Still, it outperforms several baselines having access to all views. For the pushing scenario on the other hand, we observe only a very slight drop in performance as we decrease the number of views. [Answer to iBq4]
- We added an experiment where the trained policy of the pushing scenario is applied to scenes that contain more than one object. No retraining was necessary to achieve these results. NeRF-RL outperforms the baselines in this scenario as well. [Answer to aWEL]
- We added an experiment investigating a baseline using a different choice of keypoints (more) in the mug environment. The performance of this baseline is slightly better than the performance of the keypoints baseline shown in the original paper, but is still surpassed by compositional NeRF-RL. [Answer to iBq4]
- We clarified how we chose the view for the CURL baseline. [Answer to iBq4]
- Along with the experiments mentioned above, we added a discussion of the data coverage required for NeRF-RL. [Answer to iBq4, aWEL]
- Along with the 2D experiments mentioned above, we discuss the applicability of our method in 2D scenarios. [Answer to iBq4, aWEL]

---

> ### Author Response · Authors · 2022-08-09
> **Authors Second Response - Overview**
>
> We would like to thank the reviewers for their constructive responses to our initial answers and updates. In response to a follow up question of Rev. iBq4, we have added two more experiments investigating the keypoints baselines, and updated the paper accordingly.

---

### Meta-Review · Area_Chair_o2NF · 2022-08-27

**Recommendation:** Accept
**Confidence:** Certain

**Metareview:**

The reviewers appreciated the detailed replies and the additional experiments. They are now all in favor of publishing the paper.
The paper proposes to use the NeRF representation on RL, the benefits of this combination is shown in a wide variety of experiments. The last open points by iBq4 and aWEL were well addressed by author replies and additional experiments.

**Award:**

Yes

---

### Decision · Program_Chairs · 2022-09-14

Accept